# Tourist Experience Challenges: A Holistic Approach

Virginica Rusu [1,*], Cristian Rusu [2,*] , Nicolás Matus [2,*] and Federico Botella [3]

1  Departamento de Humanidades, Universidad de Playa Ancha de Ciencias de la Educación,
   Valparaíso 2340000, Chile
2  Escuela de Ingeniería Informática, Pontificia Universidad Católica de Valparaíso, Valparaíso 2340000, Chile
3  Instituto Centro de Investigación Operativa, Universidad Miguel Hernández de Elche,
   Avenida de la Universidad s/n, 03202 Elche, Spain; federico@umh.es
*  Correspondence: virginica.rusu@upla.cl (V.R.); cristian.rusu@pucv.cl (C.R.);
   nicolas.matus.p@mail.pucv.cl (N.M.)

**Abstract:** Tourist experience (TX) has been covered by many studies. However, a consensus on the topic still needs to be reached in terms of its dimensions, factors, evaluation methods, and evaluation models. Moreover, the COVID-19 pandemic severely affected the tourism sector, and the post-pandemic era could bring about new challenges and opportunities, such as the growing awareness of the need for greener, more sustainable, and more balanced tourism. In this study, we consider TX a particular case of customer experience (CX) and an extension of the user experience (UX) concept. We conducted a systematic literature review addressing the concept of TX and reviewing articles published from 2012 to April 2023, indexed in two significant and relevant databases (Web of Sciences and Science Direct). We addressed research questions concerning (1) TX definition; (2) TX dimensions, attributes, and factors; (3) methods used to evaluate TX; and (4) the post-pandemic TX. We selected and thoroughly analyzed 167 articles. We analyze the TX concept, models, evaluation, and the post-pandemic context. We propose a holistic definition of TX and recommend ways to achieve its better analysis. Lessons learned during the COVID-19 pandemic may be helpful when dealing with future challenges and crises.

**Keywords:** tourist experience; customer experience; tourist experience model; tourist experience evaluation; COVID-19 post-pandemic tourist experience

## 1. Introduction

The World Tourism Organization (UNWTO) defines tourism as "a social, cultural and economic phenomenon which entails the movement of people to countries or places outside their usual environment for personal or business/professional purposes" [1]. Tourism is an intangible commodity that is continuously evolving. Virtual channels replace traditional channels for tourism promotion/sales/feedback. The global COVID-19 pandemic dramatically affected tourism, which at one point became almost impossible, with virtual tourism being the only option. In May 2023, the World Health Organization (WHO) declared that COVID-19 was no longer a public health emergency of international concern [2]. The WHO indicated that this "does not mean the pandemic itself is over, but the global emergency it has caused is, for now". The tourism sector seems to be fully recovering to its pre-pandemic levels. However, new challenges are becoming evident, such as overcrowding and the need for more sustainable tourism. Lessons learned during the pandemic may serve when dealing with future crises.

Many authors have discussed tourist experience (TX) as a concept. However, there still needs to be more consensus on its definition for several reasons, namely the subjectivity of the issue, how personal the tourist experience can be, and how multidimensional it is [3]. Scholars agree that TX spans across a period that begins when the tourist is planning a trip and continues even after this trip occurs through memories about the acquired activities

and learning. Tourism is an intangible commodity, and TX is the result of accumulated experiences and memories related to travel [4].

Customer experience (CX) is a key concept in service science. It is highly interdisciplinary; originally proposed in marketing, it has lately raised interest in several fields. Laming and Mason (2014) highlight that CX includes a customer's responses before, during, and after interacting with a brand/company during their whole "journey" [5]. TX may be considered a particular case of CX. Tourists are specific types of customers that use tourism-related services, products, and systems. However, general CX definitions do not cover the specificity of TX. We examined TX as a particular case of CX in previous works, identifying scenarios and touchpoints [6–8]. TX is strongly related (but not limited) to the services' quality.

CX can be considered an extension of user experience (UX), a well-known and highly explored topic in human–computer interaction (HCI) [9]. CX focuses on a person's interaction with all services, systems, and products that a company/organization/brand offers instead of focusing on the interaction with a single product, system, or service, as UX does. The ISO 9241-210 standard defines UX as "the perceptions and responses of the person resulting from the use and/or anticipated use of a product, system, or service" [10]. The definition covers software systems and can be applied to tourism-related products, systems, and services. We think that UX is an excellent approach to TX. We have focused on UX tourism-related digital products, especially online travel agencies [11–14], virtual museums [15–17], and national parks [18]. We have also used a broader, holistic approach to TX from a CX point of view [3,19–21].

Companies are increasingly aware that CX plays a key role in determining success. A good CX may increase customer attraction and retention. CX evaluation is therefore critical in order to understand how customers perceive a company/organization and its products/systems/services. We think that assessing TX, as well as CX, is challenging for several reasons: TX is constructed through touchpoints of different natures, and it is likely that the experience at one touchpoint may (highly) influence experiences at subsequent touchpoints; TX is multidimensional; and TX is highly personal [13]. In order to properly design and evaluate TX, we must first understand what TX is and which TX dimensions/factors/attributes are relevant.

Relevant research has been carried out on TX, especially over the last decade. However, there is still no agreement on the concept, its dimensions, and evaluation. Moreover, despite the huge impact of the COVID-19 pandemic on tourism, studies on TX and COVID-19 are still scarce and limited to specific contexts. In order to identify the research trends and gaps, we have performed a systematic literature review of studies published from 2012 to 2022 (until April 2023). The objectives of our study are to identify and in-depth analyze (1) TX definitions, (2) TX dimensions, (3) TX evaluation methods, and (4) the COVID-19 pandemic TX. We have focused our search on two significant databases: Science Direct and Web of Science. We have used a holistic approach to TX, which we consider a particular case of CX and a highly interdisciplinary topic. We found several TX definitions, and we propose a holistic definition of TX. Scholars propose a variety of TX dimensions; this confirms its multidimensional nature. The results show that most studies use scales when evaluating TX and its antecedents and consequences. Most of the studies utilize a quantitative approach to assess TX. However, holistic TX evaluations should focus on all (or at least the most relevant) touchpoints, dimensions, and context. Even if COVID-19 is no longer a global emergency, several lessons learned during the pandemic may be helpful when dealing with future challenges and crises.

## 2. Background

### 2.1. User Experience

UX is a well-known concept in HCI. The Association for Computing Machinery (ACM) Special Interest Group on Computer–Human Interaction (SIGCHI) declares its scope to be "the study and practice of the design, implementation, use, and evaluation of interactive

computing systems" [22]. However, the UX definition in ISO 9241-210 standard indicates that UX is not restricted to the use of interactive software systems but applies to any product, system, or service [10]. Therefore, the UX concept is also useful when studying tourists' interactions with tourism-related products, systems, or services. In tourism, UX refers to tourists' emotions, expectations, responses, and behaviors before, during, and after they use such products, systems, or services.

### 2.2. Service Science and Customer Experience

Maglio and Spohrer (2008) consider service science to be "the study of service systems, which are dynamic value co-creation configurations of resources (people, technology, organizations, and shared information)" [23]. Service science is a highly interdisciplinary field that combines organizational, human, business, and technological understanding, to explain service systems, including how they interact and evolve to co-create value [24]. Schmitt (1999) proposes a shift from traditional marketing to experiential marketing as "products, communications, and marketing campaigns that awaken senses and reach the heart of the consumer, that they manage to generate a type of stimulation in people" [25]. He identified a set of strategic experiential modules: sensory experiences, affective experiences, cognitive experiences, physical experiences, behaviors and lifestyle, and social-identify experiences.

CX has become a key concept in service science in recent decades. Several CX definitions have been proposed, but no agreement exists on the CX concept. Laming and Mason (2014) indicate that CX includes "the physical and emotional experiences occurring through the interactions with the product and/or service offering of a brand from the point of first direct, conscious contact, through the total journey to the post-consumption stage" [5]. Joshi (2014) also stresses that CX is a sum of experiences when there is a service relationship between the customer and the company [26]. LaSalle and Britton (2003) have a similar approach to CX, which in their view, is generated by the set of interactions between the customer and a product, a company, or part of its organization, which provoke customers' reactions [27].

Meyer and Schwager (2007) indicate that CX is built through "touchpoints", the points in time when customers are "touching" any product, service, or system that a brand offers across multiple channels, and experiences are generated [28]. Stein and Ramaseshan (2016) have identified seven types of elements present at touchpoints: atmospheric, technological, communicative, process, employee–customer interaction, customer–customer interaction, and product interaction [29]. Vanharanta et al. (2015) emphasize that CX is a holistic concept that involves subjective experiences and includes customers' rational thoughts and emotions [30]. Scholars have proposed several CX models. Gentile et al. (2007) synthesized a comprehensive set of CX dimensions: sensorial, emotional, cognitive, pragmatic, lifestyle, and relational [31].

We agree with Lewis's approach to CX as an extension of the UX concept [9]. In our view, UX focuses on a person's interactions with a single product, system, or service; CX offers a holistic approach, focusing on a person's interactions with all products, systems, or services that a company (institution, brand) offers throughout their whole "journey" as a customer. CX is undoubtedly more complex than UX.

### 2.3. Tourism Experience

There still needs to be a consensus on TX definition, dimensions, evaluation, and management. TX may be considered a particular case of CX; tourists are specific types of customers who use tourism-related services, products, and systems. However, general CX definitions may not cover TX specificity.

A wide range of TX dimensions have been identified by different authors. Most of them are related to CX models. Pine and Gilmore (1998) identified four dimensions ("realms") of experiences: entertainment, educational, escapist, and esthetic [32]. Their "4Es" model is still used in many TX studies. They argued that the economy had been switched from a service-based approach to an experience-based one. They have also

pointed out the need to design "memorable" experiences that will be remembered and recalled after the event has occurred. This later defined the memorable tourist experience (MTE) concept. Instead of MTE, we will consistently use the abbreviation MTX to highlight the strong relationship between TX (and its variants, as MTX), CX, and UX. Walls et al. (2011) proposed a "framework of factors" that influence TX: individual characteristics, perceived human interactions in the destination, and situational factors [33].

Ritchie and Hudson (2009) identified six "streams" in TX research: conceptual, oriented to tourist behavior; methodological, oriented to specific kinds of tourism; and managerial, oriented to levels/types of experiences [34]. Understanding TX and its dimensions/factors is critical to evaluating, designing, and managing TX. Thompson (2018) indicated some key issues in CX research: underestimating qualitative research and emphasizing only quantitative research, correctly integrating and interpreting data, and the disconnection between evaluation and design [35]. We think that the same challenges are also present in TX research.

## 3. Research Method

This systematic literature review (SLR) was performed guided by the framework for literature review proposed by Kitchenham (2004) [36], which includes three stages: (1) planning the review, (2) conducting the review, and (3) reporting the review. Additionally, we have incorporated the PRISMA methodology checklist into our review [37].

### 3.1. Research Questions

We focused our study on (1) TX definitions, (2) dimensions and factors regarding TX, (3) methods that are used to evaluate TX, and (4) the COVID-19 (post-)pandemic TX context. Table 1 presents the four research questions (RQ) that guided our study.

**Table 1.** SLR Research Questions.

| ID | Research Question |
|---|---|
| RQ1 | What is TX? |
| RQ2 | What are the TX dimensions and what factors influence TX? |
| RQ3 | What methods are used to evaluate TX? |
| RQ4 | How is the post-pandemic TX? |

### 3.2. Literature Search

We examined the literature published within the last decade (from 2012 to April 2023) and indexed in two databases: Web of Science (WOS) and Science Direct (SD). We used the search string '"tourism experience" OR "tourist experience" OR "touristic experience"'. Initially, we also searched for studies indexed in other relevant databases, such as Scopus. However, we later decided to focus only on WOS and SD for two reasons: (1) to keep information manageable, as, for instance, in Scopus, we found almost 4000 studies; and (2) most of the studies indexed in WOS and SD were published in journals and were submitted to a rigorous reviewing process. Both the number and the percentage of studies available in each database are listed in Table 2.

**Table 2.** SLR results by database.

| Database | Number of Studies | % of Studies |
|---|---|---|
| Web of Science | 1399 | 69.6% |
| Science Direct | 610 | 30.4% |
| Total | 2009 | 100% |

*3.3. Study Selection*

The inclusion and exclusion criteria are presented in Table 3.

**Table 3.** SLR inclusion and exclusion criteria.

| ID | Category | Criteria |
|---|---|---|
| IN1 | Inclusion | Articles published in the last decade (from 2012 to April 2023) |
| IN2 | Inclusion | Articles referring to at least one research question |
| EX1 | Exclusion | Remove duplicated articles |

We gathered 167 articles for a full review by applying the selection criteria. Figure 1 shows the search and selection process flow, using the inclusion and exclusion criteria and removing duplicates. Only articles in English were reviewed.

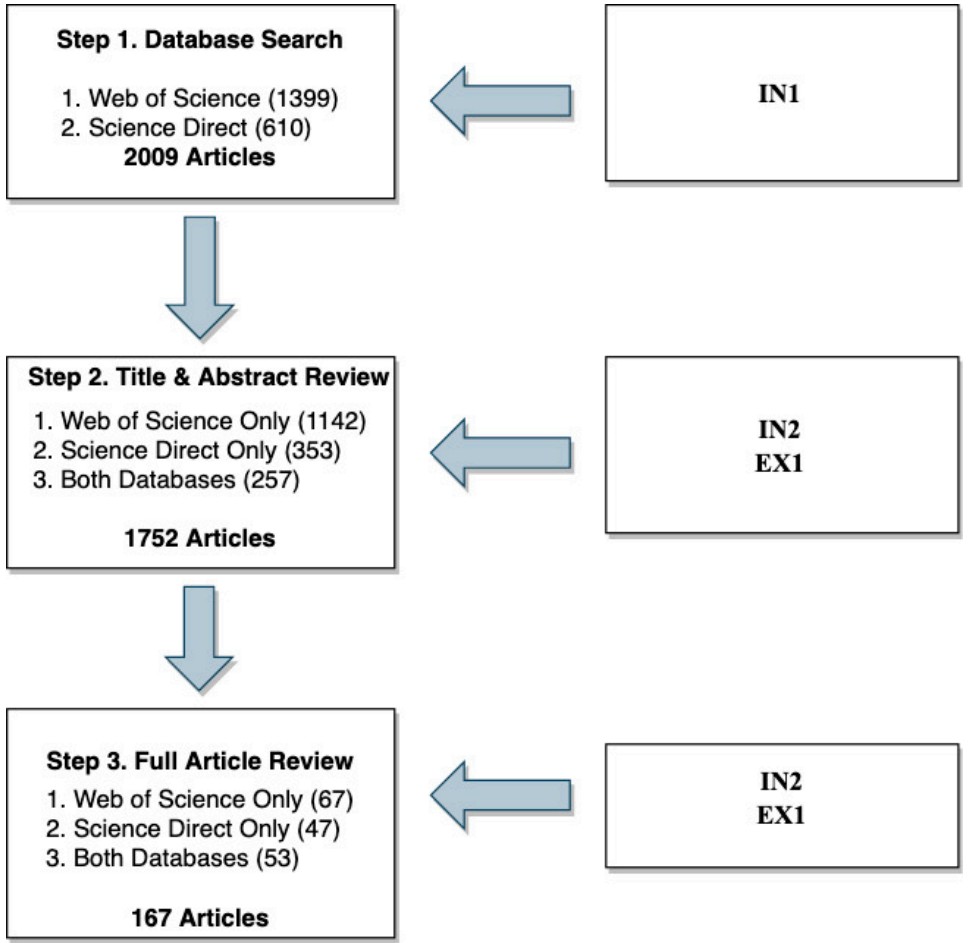

**Figure 1.** Flow chart with the results of the study selection process.

## 4. Data Synthesis

This section includes the data synthesis for the selected studies, identifying (i) publications over the years, (ii) document type and associated subject area, and (iii) distribution of selected articles by database.

*4.1. Year of Publication*

Figure 2 indicates an increasing interest in TX research. Information regarding 2023 is highlighted in a distinct color, as it was collected in April 2023; it is very likely that the total number of studies to be published in 2023 will be significantly higher.

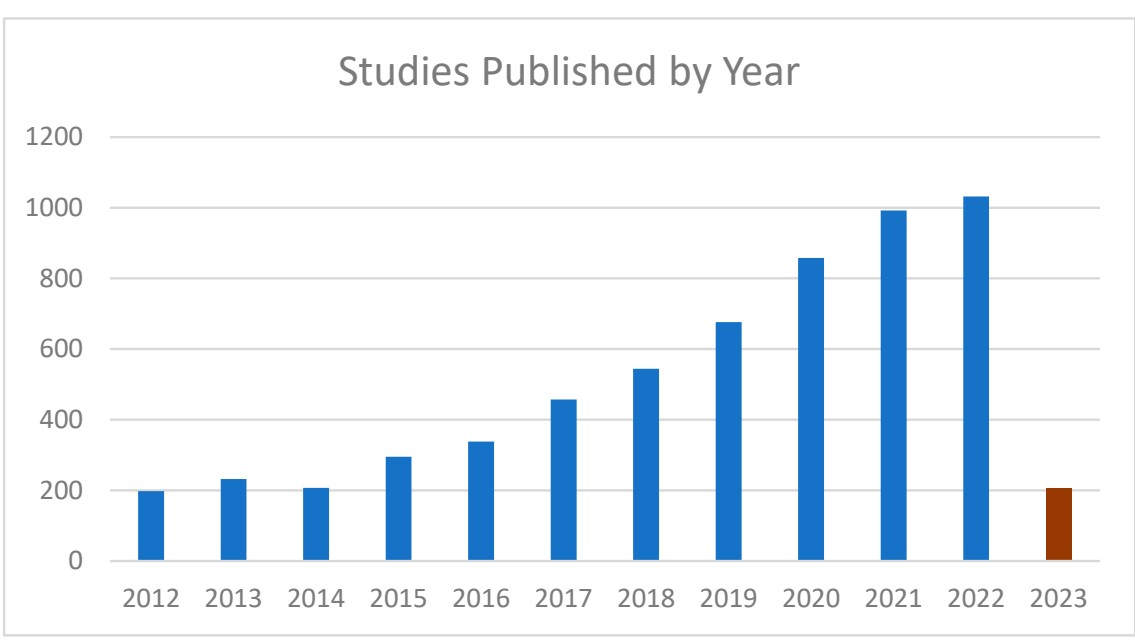

**Figure 2.** Research on the TX area over the years.

*4.2. Total Studies by Database*

As seen in Figure 3, of the 1752 articles identified based on the search string (after removing duplicated), 65.2% were indexed only in WOS, 20.1% were indexed only in SD, and 14.7% were indexed on both databases.

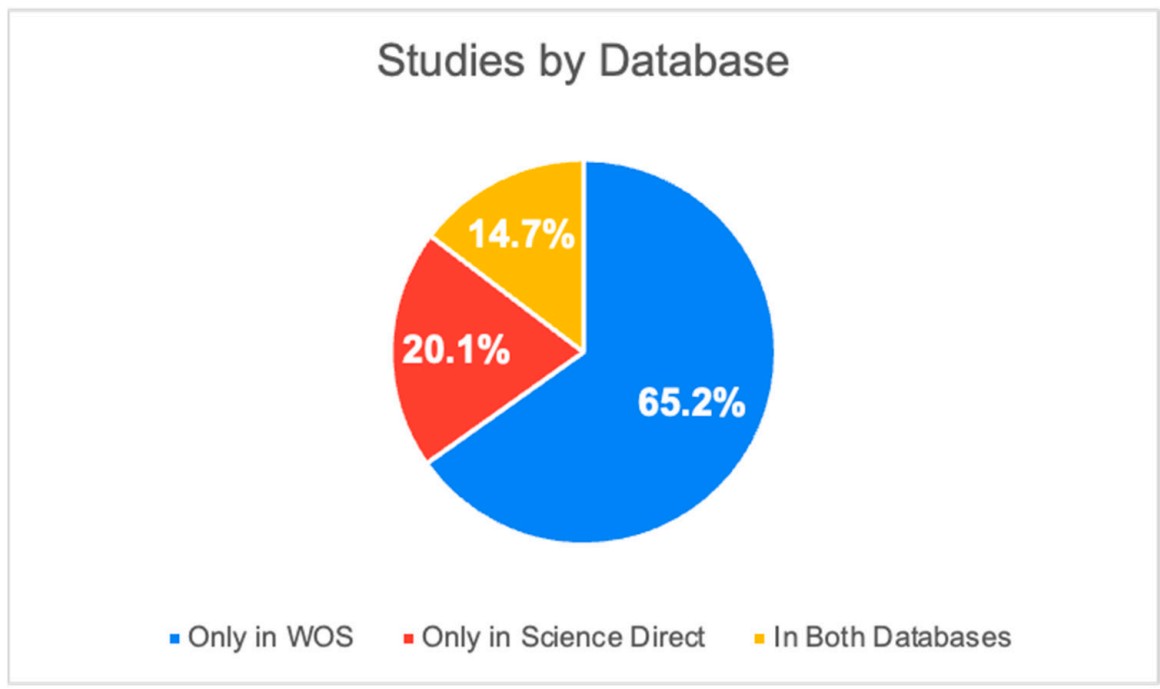

**Figure 3.** Studies number per database.

*4.3. Document Type and Subject Area*

Figure 4 shows that most documents correspond to articles: 1319 in Web of Science and 551 in SD. Classification areas differ in WOS and SD, but most are associated with hospitality, business and management, and social sciences.

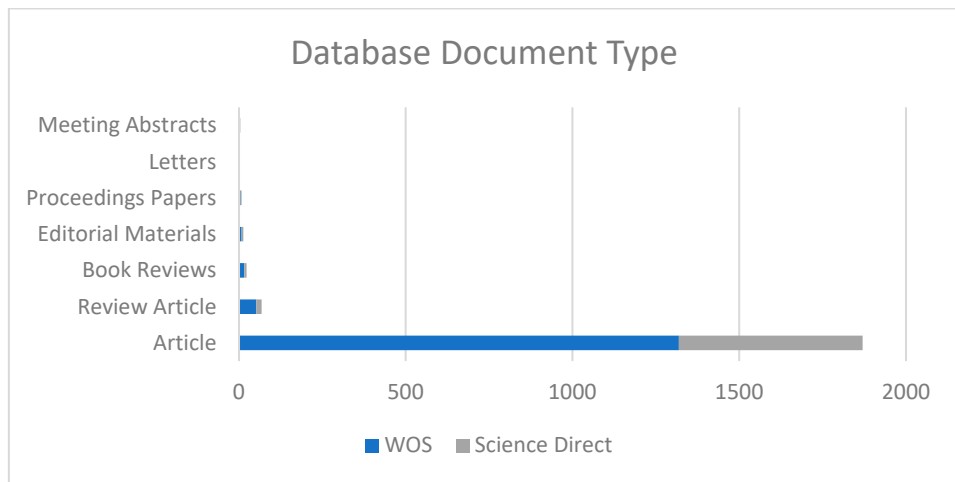

**Figure 4.** Document type by database.

*4.4. Selected Studies by Database*

As seen in Figure 5, of the 167 selected studies for full review, 40.1% were indexed only in WOS (67), 28.1% were indexed only in SD (47), and 31.8% were indexed in both databases (53). Compared to the first step of the selection process, the percentage of studies selected in step 3 and indexed in both databases increased, and the distribution of studies by database was more balanced.

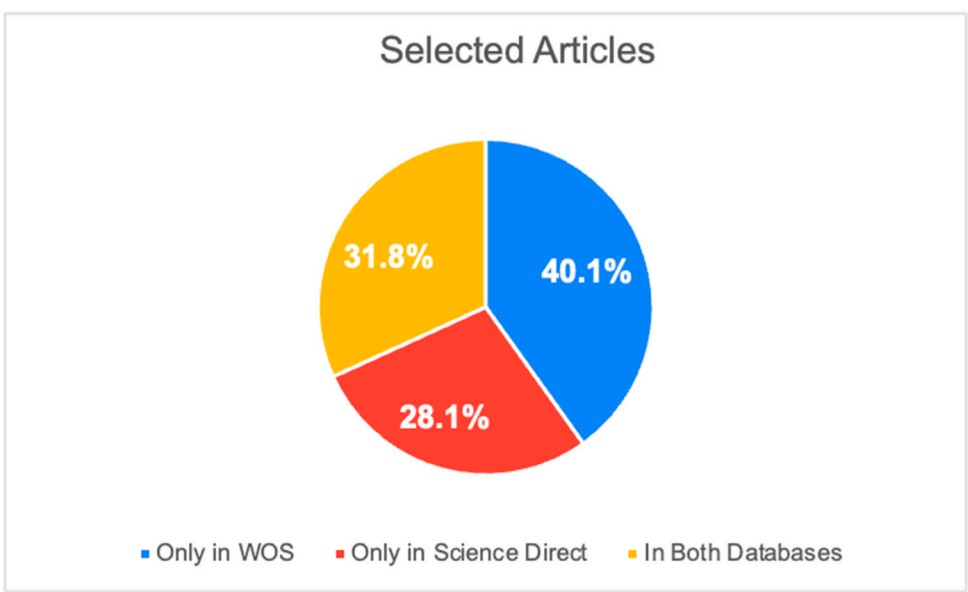

**Figure 5.** Selected articles by database.

## 5. Research Questions

*5.1. RQ1: What Is TX?*

Godovykh and Tasci (2020) highlight the need for a consensus on the CX concept [38]. They propose a holistic definition of experience: "the totality of cognitive, affective, sensory, and conative responses, on a spectrum of negative to positive, evoked by all stimuli encountered in pre, during, and post phases of consumption affected by situational and brand-related factors filtered through personal differences of consumers, eventually resulting in differential outcomes related to consumers and brands". Even if their definition refers to experiences in tourism and aims to include all TX-relevant aspects, it does not limit to TX. It may apply to experiences in other fields.



Sugathan and Ranjan (2019) indicate several reasons why experiences in tourism differ from consumption experiences in general [39]: (1) tourism services are highly experiential, and their utility is not readily observable; (2) tourism companies presume that they have the expertise to maximize tourists' experiences; (3) TX occurs in distinct stages (planning the event, experiencing the destination, and constructing narratives after returning); (4) co-creation can occasionally impede value creation, leading to dampening of TX; and (5) novelty is a primary motive for tourists, and they may not want to return to the same place, even when having good TX. Mendes et al. (2022) examine tourism as a service to enhance TX [40]. Jensen et al. (2015) argue that TX is about how meaning is created and highlight that meaning creation includes individual, social, and cultural meanings [41]. Tregua et al. (2020) identify five perspectives on TX literature [42]: firm-inspired, technology-based, cultural-based, innovation-based, and customer satisfaction and participation.

Kim et al. (2012) consider that a positive, memorable tourism experience (MTX) is "a tourism experience positively remembered and recalled after the event has occurred" [43]. Kim (2014) highlights that tourists' experiences at the destination determine their satisfaction and may generate MTX [44]. In their systematic literature review on MTX, Hosseini et al. (2021) cite its "initial definition" given by Kim et al. (2012) [45]. They highlight that not all experiences are memorable. Chen et al. (2020) show in their review that most of the MTX literature published from 2012 to 2021 focuses on MTX and tourist behavior and the Kim et al. (2012) MTX scale. Earlier studies focused more on TX, while later studies focused more on satisfaction, marketing, and service innovation. They think that MTX research should use a more customer-centered approach.

Servidio and Ruffolo (2016) described MTX as "an important event stored in the memory and recalled after it has occurred" [46]. They indicate that memory is partially neglected in TX studies. Zhang et al. (2018) highlight that MTX and TX are related yet different concepts "in connotation and extension"; not all TXs convert into MTXs [47]. Bigne et al. (2020) compare MTX with "ordinary" TX (OTX) [48]. They consider the non-MTX as ordinary OTX. Anaya and Lehto (2023) focus on memorable tourism moments (MTMs) [49]. Coelho et al. (2018) attempt to identify the core processes that are sense-making and meaningful in MTX [50]. They highlight that studies on MTX have three perspectives: expansive, managerial/economic, and modeling. They conclude that it may be more enriching to identify MTX components rather than focusing on what makes an experience memorable. Park and Santos (2016) highlight that what tourists most recall in the post-travel stage are unique and unexpected personal experiences [51].

Lu et al. (2015) focus on tourists' satisfaction to assess TX [52]. Sarra et al. (2015) think that the future of any tourist product and service depends on the perceived quality of TX [53]. Altunel and Erkut (2015) consider TX to be the quality of experience concerning tourism [54]. On the contrary, Fernandes and Cruz (2016) consider experience as "a far broader and less delimited concept than product or service quality", and they think a holistic approach to TX is still missing [55].

Brunner–Sperdin et al. (2012) indicate that scholars focus on cognitive components, neglecting emotional aspects of customer satisfaction [56]. Fall Diallo et al. (2022) emphasize that TX is highly personal [57]. Volo (2021) highlights the role of emotions in tourists' cognitive evaluations and behavioral responses [58]. Lin and Kuo (2016) identify the psychological process through which TX generates perceived value, which creates satisfaction, and satisfaction is a dominant antecedent of loyalty intentions [59]. daSilva et al. (2021) synthesize the subjective optimal experience (flow) in TX [60]. Nawijn and Biran (2018) point out that TX research should also focus on negative emotions, not only on positive emotions [61]. Meacci and Liberatore (2018) highlight the role of the senses in TX [62]. They argue that each touchpoint of TX has a specific sensory dimension, and a holistic multisensorial approach should be applied when designing TX.

Wang et al. (2014) emphasize that travel is a process that includes three stages: pre-trip, during-trip, and post-trip [63]. Stienmetz et al. (2021) explore the complex nature of TX,

which occurs through discrete (moment-to-moment) yet summarized events [64]. Björk et al. (2020) highlight that TX has been lately one of the most addressed topics of Nordic TX research and emphasize the 3 phases of travel [65]. Aybek and Ozdemir (2022) indicate that the ethnic restaurant experience should be considered a processual activity [66]. Unger et al. (2016) analyze the business travel experience in 4 phases: trip preparations, passenger experience, destination experience, and homecoming [67].

Teoh et al. (2021) acknowledge that TX can have transformative attributes, identifying three dimensions of transformative TX: experience, experience-facilitator, and experience-consumer [68]. Inversini et al. (2022) highlight the transformative power of TX and its potential to generate significant change and impact on individuals' self while on vacation [69]. Chang et al. (2021) focus on the creative aspects of experiences in tourism [70]. Cao et al. (2023) stress the role of co-creation in TX [71]. Campos et al. (2015) propose a psychology-based definition of on-site co-creation TX [72]. Yamashita (2015) argues that the "authentic" TX is impossible nowadays, as, in the pre-travel stage, tourists already collect information on what they will experience [73]. Verma et al. (2022) see virtual tourism as "the future of tourism", revolutionizing TX without traveling [74]. Huang and Choi (2019) focus on tourist engagement [75]. Weaver et al. (2017) also stress the importance of effective engagement [76].

Koç et al. (2022) point out that other tourists can influence one's TX positively through co-creation or negatively through co-destruction [77]. Uysal et al. (2016) indicate that the impact of TX on the quality of life depends on stages in life and other background variables that influence the degree of importance of travel [78]. Miyakawa and Oguchi (2022) examine the family TX [79]. They stress the beneficial effects of family tourism on parents' well-being and their children's generic skills.

Tan (2017) highlights the link between the physical and virtual spaces connected through the smartphone [80]. Kabadayi et al. (2021) indicate that smart services can improve TX [81]. Femenia-Serra and Neuhofer (2018) conclude that experiences in smart destinations are data-driven, built-in in real-time, based on context awareness, and co-created [82].

Sedgley et al. (2017) study the TX of mothers of children with autism spectrum disorder (ASD) [83]. They analyze the challenges of caring for children with ASD on holiday and the perceived benefits of holidays. Olson and Reddy-Best (2019) highlight that transgender and gender non-conforming individuals often feel fear and anxiety about sharing their gender identity in identification, security thresholds, and check-in procedures [84].

Dodds (2020) proposes a model of TX's "lifecycle", indicating that a tourist may transit through four stages: excitement, novelty, normalization, and familiarity [85]. Cajiao et al. (2022) identify four motivations of Antarctic tourists: experience and learning, adventure, social bonding, and the lifetime trip [86]. Pafi et al. (2020) argue that TX of coastal landscapes needs to be understood from a community-led rather than market-led perspective [87]. Suhartanto et al. (2021) indicate the need to examine TX holistically in halal tourism, including leisure experience and fulfilling religious requirements [88].

Tussyadiah (2014) suggests three fundamentals in TX design [89]: human-centeredness, iterative designing process, and a holistic experience concept as an outcome of designing. Eide et al. (2017) insist on maintaining innovation over time when managing TX [90]. They refer to the "maintenance of experience concept innovation", as keeping the experience attractive and "alive" over time. Soler and Gemar (2017) conclude that tourism management is responsible for TX quality [91]. Van der Zee et al. (2017) recognize that a "network approach" in tourism management is likely to improve TX [92]. Ferguson et al. (2017) indicate that tipping or non-tipping behavior predictor could enable better TX management [93].

Table 4 synthesizes the approaches to the TX concepts and some of the studies associated with each approach.

**Table 4.** Approaches to the TX concept.

| Approach | Studies |
| --- | --- |
| Conceptual | [38–42] |
| Memorable | [43–51] |
| Satisfaction | [52–55] |
| Psychology, Senses, and Emotions | [56–62] |
| Process | [63–67] |
| Transformative | [68,69] |
| Creative | [70–74] |
| Engagement | [75,76] |
| Influences | [77–82] |
| Special needs and Tourist segments | [83–88] |
| Management | [89–93] |

In our view, the definition of Godovykh and Tasci is holistic and attends to most of the concerns expressed by Sugathan and Ranjan. It also follows at least two of the TX perspective identified by Tregua et al., i.e., brand-related and customer-related. The definition refers to a brand, but a trip usually involves several brands (companies, organizations). We think TX may be referred to as the cognitive, affective, sensory, and conative subjective perceptions, either negative or positive, and affected by situational factors, that a tourist has when interacting with brands, pre-, during, and post-travel, including their outcomes.

The focus on MTX in many studies is remarkable—however, TX range from negative to positive. Considering the OTX as the opposite of MTX normalizes the possible (very) negative TX. TX should have a human-centered approach and attend to the whole range of experiences, negative, ordinary, and memorable. The link between service quality, satisfaction, and TX is obvious and empirically confirmed by many studies. We agree that TX is more than service quality and satisfaction; a holistic approach to TX is necessary. Conceptual research is essential, but all assumptions should be tested empirically. The emotional aspects of TX are certainly equally crucial as the cognitive ones. A holistic approach to TX should be multisensorial and include positive and negative emotions.

The impact of co-creation and tourist engagement in TX was emphasized in several studies, and the search for authenticity is recurrent when studying tourist motivation. However, opinions range from considering authentic TX as impossible nowadays to assuming that virtual tourism is the future. The effect of tourist-to-tourist interactions on TX has been recognized as negative or positive (in the case of family tourism). The authors agree on the highly personal and subjective nature of TX, but only some studies focus on tourists with special needs.

Many authors highlight that TX is constructed through three stages (pre-, during, and post-travel) and stress the importance of analyzing the whole process. This is consistent with the CX approach to TX, where the moment-to-moment events proposed by Stienmetz et al. are, in fact, touchpoints where tourist interacts with tourism-related products, systems, or services. Several studies focus on the positive impact of technologies on TX: smartphones, smart services, and smart destinations. However, they are not linking UX and TX. This link is necessary for a couple of reasons: (1) CX and TX are extensions of UX, and (2) the use of technologies in tourism is continuously increasing. Therefore, even if UX were to focus only on tourists' perception of "smart" tourism, its relationship with TX is undeniable.

*5.2. RQ2: What Are the TX Dimensions and What Factors Influence TX?*

Fernades and Cruz (2016) indicate the lack of research on CX "measurement", and its underlying dimensions; in their view, CX dimensions have only been assumed but not yet "extracted" [55]. Godovykh and Tasci (2020) identify the four most frequently explained components of experience: affective, cognitive, conative, and sensorial [38]. They indicate that scholars use different terminology when referring to the same concept. Chen et al. (2020) use four MTX stimuli (psychological experience factors): hedonism, novelty,

meaningfulness, and social interaction: [94]. Moliner et al. (2019) propose a TX model with five dimensions: cognitive, affective, behavioral, sensory, and social [95]. Su et al. (2022) use seven dimensions of TX: sensory, feeling-based, behavioral, knowledge, authenticity, co-creation, and novelty [96].

Escobar et al. (2019) identify 11 factors of delightful TX that they group into two dimensions: cognitive and affective [97]. Huang and Liu (2018) consider four dimensions of the effects of a creative experience: peace of mind, escape, unique involvement, and interactive opportunities [98]. Zatori et al. (2018) identify four dimensions of on-site TX [99]: emotional, mental, flow-like, and social experience–involvement. Chen et al. (2022) consider four TX dimensions: sensory experience, action experience, emotional experience, and thinking experience [100].

Kim et al. (2012) have developed a memorable tourist experience (MTX) scale that includes seven dimensions [43]: hedonism, refreshment, local culture, meaningfulness, knowledge, involvement, and novelty. The dimensions that they have proposed were used by many other scholars when developing TX(-related) scales. Kim (2014) identifies 10 dimensions that affect MTX [44]. Bigne et al. (2020) synthesize eight MTX dimensions based on other scholars' studies [48]. Luo et al. (2020) propose seven dimensions for entertainment TX [101]. Coelho et al. (2018) acknowledge that Kim et al. developed one of the most widely used MTX, but they propose only three MTX dimensions: personal, relational, and environmental. Hosseini et al. (2021) synthetize the MTX dimensions mentioned in literature from 2012 to 2021 [45]. They find that the most recurrent dimensions from 2012 to 2014 are the ones that Kim et al. use in their MTX scale. They indicate that, from 2015, new dimensions such as behavioral intention, destination image, loyalty, satisfaction, word-of-mouth (WOM) intention, and revisit intention have been studied. They pointed out that starting in 2018, most studies began to focus on antecedents and consequences of MTXs and consider the MTX scale dimensions to be interdependent; new factors were considered, such as the environment, destination attributes, and personal and social considerations.

As proposed by the studies under review, TX dimensions are synthesized in Table 5.

**Table 5.** TX dimensions.

| Dimensions | Studies |
| --- | --- |
| Affective | [38,95–97,99,100] |
| Cognitive | [38,95,97] |
| Conative | [38] |
| Sensorial | [38,95,96,100] |
| Behavioral | [45,95,96] |
| Social | [95,99] |
| Knowledge | [43,45,48,96] |
| Authenticity, Co-creation | [94,96] |
| Novelty | [43,45,48,96,101] |
| Peace of mind, Interactive opportunities | [98] |
| Escape | [98,101] |
| Involvement | [43,45,48,98,101] |
| Mental, Flow-like | [99] |
| Action, Thinking | [100] |
| Hedonism, Meaningfulness | [43,45,48,94] |
| Refreshment | [43,45,48,101] |
| Local culture | [43–45,48,101] |
| Environment management | [44,50] |
| Variety of activities, Hospitality, Infrastructure, Accessibility, Quality of service, Physiography, Place attachment, Superstructure | [44] |
| Serendipity and surprises | [48] |
| Learning, Enjoyment | [101] |
| Personal, Relational | [50] |
| Destination image, Loyalty, Satisfaction, Word-of-mouth intention, Revisit intention | [45] |

Several studies analyze the antecedents and consequences of TX. Zhang et al. (2018) have examined the relationship between the country and destination image, MTX, and

revisit intention [47]. In their study on the perceived coolness of "Generation Y" (born between 1977 and 1994) in the context of creative tourism, Chen and Chou (2019) propose a model of creative TX [102]. The model includes three antecedents and three consequences of perceived coolness. Kim et al. (2021) study the impact of mobility on MTX [103]. Ruiz et al. (2021) evaluate the effect of crowding on tourist satisfaction [104]. Shaykh–Baygloo (2021) examine the relationships between a sense of place, the perceived quality and value of attractions, and tourists' satisfaction [105]. Kirillova et al. (2014) indicate that the importance of the aesthetic attributes of a destination is widely accepted. Still, it is usually reduced to a single-dimensional variable such as "the place is beautiful" [106]. Koç et al. (2022) have studied the relationship between negative tourist-to-tourist interaction, tourist emotions, and intention to recommend [77].

Various studies propose specific, domain-related TX dimensions. Chang and Hung (2021) identify seven dimensions of cultural and creative tourism [107]. Gardiner et al. (2022) identify five factors ("realms") that influence the attitudes toward recreated historical tourism experiences [108]. When analyzing TX in Lisabon, Sarra et al. (2015) focus on three aspects [53]. Pafi et al. (2020) identify five types of landscape experiences in coastal tourism [87]. Sthapit et al. (2022) propose a model of memorable nature-based TX that includes five dimensions [109]. Tang et al. (2021) propose a model for urban Chinese forest recreationists with five dimensions [110]. Lin et al. (2022) analyze TX in Haizhu National Wetland Park, China, using five dimensions [111]. Chen et al. (2023) study the relationship between rural TX and tourists' post-experience green consumption intention based on the 4Es model of Pine and Gilmore [112]. Liu et al. (2023) study the well-being of TX in China based on the same four TX dimensions [113]. In their study on an island destination (Jeju Island, South Korea), Moon and Han (2018) also consider only four TX dimensions [114]. Lee and Han (2019) consider seven dimensions when proposing a scale for the low-carbon tourism experience [115]. Cajiao et al. (2022) study the Antarctic TX based on two dimensions and six sub-dimensions [86]. When looking at the emotional influences of rural travel on family cohesion, Lee and Lee (2021) used five dimensions of the MTX scale of Kim et al. [116]. Voon et al. (2017) propose nine dimensions of TX in homestays (in Malaysia) [117]. Suhartanto et al. (2020) propose five dimensions for agritourism TX [118]. Komppula et al. (2016) study TX during social holidays in Finland, identifying five factors for the social TX [119]. Olivier et al. (2022) propose a model of TX in periodic hallmark festivals that they call PHF-TX; it includes six dimensions [120]. Suhartanto et al. (2021) identify five dimensions of holistic halal TX [88]. In their study on senior tourists, Sie et al. (2021) consider four factors of MTX [121]. Martins et al. (2017) argue for the importance of the use of technology in tourism and propose a "multisensorial virtual-reality tourism model", with five stimulus-related "blocks" [122]. Table 6 shows domain-related cases of TX dimensions.

Fernades and Cruz explicitly state the need for more research on CX dimensions (and consequently on TX dimensions). We consider as "dimensions" the main components of TX and as "factors" their sub-categorizes (sub-dimensions). However, the line between dimensions/factors/attributes is quite blurred in many studies. Godovykh and Tasci identify four main TX dimensions: affective, cognitive, conative, and sensorial; however, they also indicate that these dimensions are frequently referred to under different names. The four components are recurrent in most of the TX models. Kim et al. define a set of seven MTX dimensions that have subsequently been used by other scholars: hedonism, refreshment, local culture, meaningfulness, knowledge, involvement, and novelty.

The most recurrent TX models the studies are based on are the MTX seven-dimension model of Kim et al. and the 4Es four-dimension model of Pine and Gilmore. As Hosseini et al. indicate, TX's new dimensions and factors have lately been proposed. It is worth mentioning that the same constructs are considered in some studies as TX dimensions and in others as antecedents or consequences of TX. The limit between TX, its antecedents, and its consequences is indistinct. When studying domain-related and particular cases of TX, the spectrum of dimensions and factors is very ample.

**Table 6.** Domain-related TX dimensions.

| Domain | Dimensions | Studies |
|---|---|---|
| Cultural and creative tourism | Learning, Recreation, Exhibition, Service, Food, Facilities, and Souvenirs | [107] |
| Recreated historical tourism | Education, Entertainment, Escapism, Aesthetics and Staged authenticity | [108] |
| Cultural city | Mobility, Destination image, and Level of knowledge and appreciation of tourist areas | [53] |
| Landscape experiences in costal tourism | Well-being experiences, Conscientious travel experiences, Nature experiences, Coastal change experiences, and Cultural experiences | [87] |
| Nature-based tourism | Novelty, Experiencescape, Experience co-creation, Experience intensification, and Satisfaction | [109] |
| Urban forest recreationism | Destination image, Recreational benefits, Satisfaction, Perceived value, and Behavioral intentions | [110] |
| Wetland Park | Travel cost, Escape experience, Parent–child experience, Activity experience, and the Crowded perception | [111] |
| Rural tourism | Education, Esthetic, Entertainment, and Escapism | [112] |
| Well-being | Educational, Entertainment, Esthetic, and Escapist | [113] |
| Island destination | Escapism, Relaxation, Enjoyment, and Involvement | [114] |
| Low-carbon tourism | Sensory experience, Affective experience, Learning experience, Sociocultural experience, Behavioral experience, Escapism experience, and Prestige experience | [115] |
| Antarctic tourism | Experiential (Perceived learning, Measured learning, and Satisfaction), and Pro-environmental (Environmental concerns, Management preferences, and Behavior intentions) | [86] |
| Family rural travel | Refreshment, Novelty, Arousal, Meaningfulness, and Involvement | [116] |
| Homestays | Culture, Guiding, Accommodation, Services, Food and beverages, Journey, Natural environment, Access, and Cleanliness | [117] |
| Agritourism | Uniqueness, Learning, Staff, Escape, and Peace-of-mind | [118] |
| Social tourism | Interaction, Physical environment, Content of the activities, Food and mealtime conditions, Situational factors, and Personal factors | [119] |
| Periodic hallmark festivals | Emotional value, Functional value, Monetary value, Social approval value, Self-image congruency value, and Safety value | [120] |
| Halal tourism | Halal experience (halal accommodation, and halal facility and service), and Recreation experience (people in destination, escape and refresh, and uniqueness and staff) | [88] |
| Senior tourism | Rejuvenation, Excitement, Novelty, and Local culture | [121] |
| Virtual tourism | Visual, Audition, Olfactory, Tactile, and Gustatory | [122] |

### 5.3. RQ3: What Methods Are Used to Evaluate TX?

Kim et al. (2012) developed an MTX scale that was later used and/or adapted by many other scholars [43]. The scale includes 24 items grouped into 7 dimensions. The scale's internal consistency and validity were checked in a survey of American college students and later by Kim and Ritchie (2013) in Taiwan [123]. Kim (2014) developed a scale to measure the destination attributes associated with MTX in Taiwan [44].

Anía Melón et al. (2021) used a scale that includes 48 items, 24 of which are from the MTX scale [124]. Chen et al. (2023) have used a scale with 24 items from the MTX scale, 16 items for 4 dimensions of rural TX (based on the Pine and Gilmore 4Es model), and 21 additional items [112]. Items were rated using a 7-point Likert scale in both studies. Several other studies are also based on the MTX scale but use 5-point Likert-type ratings. Coudounaris and Sthapit (2017) [125], Sthapit and Coudounaris (2017) [126], Zhong et al. (2017) [127], Gohary et al. (2018) [128], Zhang et al. (2018) [47], Sthapit et al. (2020), [129], and Lončarić et al. (2021) [130] used the 24 MTX items and from 8 to 36 additional items and aimed to evaluate antecedents and consequences of MTX. Several studies have been based on adapted versions of the MTX scale. Chen et al. (2020) used only 3 dimensions (10 items) of the scale, adding 21 other items [94]. Rasoolimanesh et al. (2021) used 23 items of the MTX scale and 12 additional items [131]. Aydin and Omuris (2020) reexamined the MTX scale of Kim et al. in the Turkish context, using only 6 MTX dimensions with 20 items [132]. Items were rated based on either 7-point or 5-point Likert scales. Some authors argue that not all MTX dimensions were significant in the context of their studies. Some results indicate that only some of the 7 MTX dimensions seem relevant, at least in particular contexts where the studies have been performed. Coelho and Gosling (2018) argue that existing MTX scales, such as the one proposed by Kim et al. (2012) and Kim and Ritchie (2014), focus on the psychological characteristics of TX and lack a more holistic view [133]. They develop an MTX scale with 41 items grouped in 10 dimensions and validate it with Brazilian tourists.

Moon and Han (2018) have used a scale to evaluate TX quality based on the 4Es model of Pine and Gilmore [114]. Escobar et al. (2019) used a scale to examine the delightful TX [97]. Mendonça-Pedro et al. (2021) explored the role of the senses, emotions, and memories on MTX [134]. Buzova et al. (2021) point out the lack of instruments for evaluating the sensory stimuli perceived by tourists and propose an index to assess the sensory destination, the "destination sensescape" [135]. Chen et al. (2021) examined how TX touchpoints affect tourists' perceived well-being [136]. Su et al. (2022) studied how TX affects tourists' subjective well-being via recollection and storytelling [96]. Adam (2020) developed a scale to evaluate the negative tourist-to-tourist interactions [137]. Koç et al. (2022) have examined the relationship between negative tourist-to-tourist interaction, tourist emotions, and intention to recommend [77]. Ruiz et al. (2021) evaluated the effect of crowding on tourist satisfaction [104]. Wang et al. (2020) have explored the relationship between environmental stimuli and TX [138]. Huang and Choi (2019) have developed a scale to evaluate tourist engagement to capture the scope of the value co-creation process and its context-dependent nature [75]. Lin and Kuo (2016) have examined the behavioral consequences of TX in Taiwanese tourist townships [59]. Zatori et al. (2018) have developed a scale to evaluate how service providers can enhance TX in city sightseeing tours [99]. Ramires et al. (2018) studied the cultural city tourism motivations relating to specific destination attributes and their satisfaction [139]. Palos-Sanchez et al. (2021) have studied the factors influencing tourist intention to visit a city [140]. Chen et al. (2022) used a TX scale to explore the cultural tourism cities [100]. Tang et al. (2021) studied urban forest recreationists in China [110]. Bichler and Pikkemaat (2021) have explored skiers' motivation factors for visiting urban destinations with a winter sports infrastructure [141]. Suhartanto et al. (2020) developed a scale for agritourism TX [118]. Magno and Cassia (2021) studied the effect of business strategies to cope with the COVID-19 crisis in agrotourism [142]. Lee and Lee (2021) examined the emotional influences of rural travel on family cohesion [116]. Castellanos-Verdugo et al. (2016) studied the ecotourist experience in a natural park [143]. Moliner et al. (2019) examined the relationship between environmental sustainability, TX, and tourist satisfaction [95]. Lee and Han (2019) proposed a scale for low-carbon TX from the perspective of nature-based tourists [115]. Wang et al. (2012) studied the relationships between TX, service quality, tourist experiences, and revisit intention in wetland parks [144]. Xu et al. (2018) studied the relationships between TX, tourism involvement, and environmentally responsible behavior in a wetland park [145]. Cajiao et al.

(2022) used a scale to study the Antarctic TX [86]. Moliner-Tena et al. (2021) examined the relationship between MTX and destination sustainability at rural and sun and beach destinations [146]. Shen et al. (2021) studied the factors important to sailing tourists to ensure satisfying marina destination experiences [147]. Huang and Liu (2018) studied the creative experience and its impact on brand image and travel benefits when visiting famous temples [98]. Chang et al. (2020) developed a scale to evaluate pilgrimage experiences [70]. Suhartanto et al. (2021) used a scale to study the holistic experience of halal tourism and its consequences to tourist satisfaction and tourist intentions [88]. Chang and Hung (2021) developed a scale to evaluate TX for cultural and creative industry parks [107]. Orth et al. (2012) studied the role of brand-related attributions in the relationships between TX and their emotional attachments to place-based brands in international wine regions [148]. Luo et al. (2020) developed a scale to evaluate entertainment TX [101]. Olivier et al. (2022) developed a scale to evaluate TX in periodic hallmark festivals in Australia (PHFs) based on their PHF-TX model [120]. Aybek and Ozdemir (2022) studied the effects of ethnic restaurant experience [66]. Tan (2017) studied the impact of the use of smartphones on TX [80]. Lee and Jan (2022) proposed a scale to evaluate the smart TX in nature-based tourism [149]. Kullada and Kurniadjie (2020) examined the influence of digital information quality in TX [150]. Kim et al. (2021) studied the use of a mobility app and its impact on how mobility is perceived in TX [103]. Shaykh-Baygloo (2021) used a scale to study the TX of international travelers to a destination [105]. Rehman et al. (2023) studied the impact of TX on the perceived price reasonableness, regenerative tourism involvement, and the moderating effects of tourist destination loyalty and destination image [151]. Bezerra and Gomes (2019) studied passengers' loyalty to an airport in the multi-airport area [152]. Chen and Chou (2019) used a scale associated with their model of creative TX for "Generation Y" [102]. Sie et al. (2021) studied senior tourists' self-determined motivations, tour preferences, memorable experiences, and subjective well-being [121]. Tan (2017) studied the repeat visitation in the coastal town [153].

Some studies are based on multiple scales. Manthiou et al. (2022) combined three TX scales: emotional tourism experience, on-site tourism experience, and experience economy [154]. They present a composite TX scale with five dimensions: reverie, tourismscape, contemplation, transformation, and amusement. The study shows that the combined five-dimensional scale evaluates TX better than the three individual scales, but the proposed scale has yet to be described explicitly. Barnes et al. (2016) studied the impact of remembered TX in a safari park [155]. They used three questionnaires: at the entrance to the park, after visiting the park, and six weeks later. The study does not present the scale's items (specific attractions of the safari park). When examining the family TX, Miyakawa and Oguchi (2022) used pre-travel and post-travel scales [79]. Both scales included items to evaluate five well-being domains. MTX was assessed on the post-travel scale. Children's generic skills were evaluated based on their parents' ratings on both pre- and post-travel scales. Children's responses were evaluated on pre- and post-travel scales, based on the same generic skills used to capture their parents' perception. Each item was adapted by translating it into wording that children could understand; children rated items based on emoji scales and accompanied by word labels ranging from strongly disagree to strongly agree.

TX scales are detailed in Appendix A, Table A1. Most of the studies were performed in China and Taiwan (over 16% each), Spain (over 10%), and the USA (almost 6%).

Many studies are based on interviews. Pomfret (2012) examined TX in adventure activities during packaged mountaineering holidays [156]. Tan et al. (2014) studied creative tourists, active co-creators of their experiences [157]. Knobloch et al. (2016) studied the nature of individual experiences regarding personal outcomes, emotions, and meanings [158]. Anaya and Lehto (2023) studied memorable tourism moments (MTMs) [49]. Unger et al. (2016) studied the business travel experience [67]. Gao and Kerstetter (2018) studied the emotion regulation strategies tourists use during vacations [159]. Wassler and Kirillova (2019) examined the host–guest relationship in tourism as "Gazers" and "Gazees" [160]. Zhang et al. 2021 studied how the COVID-19 pandemic changed social interaction [161].

Kastenholz et al. (2012) studied rural TX in a historical village [162]. Kirillova et al. (2014) studied the dimensions of tourist aesthetic judgment for nature-based and urban tourist destinations [106]. Choi and Wong (2018) studied the negative impact of confusing toponymy and place name conversion on foreign tourists' experiences [163]. Ebejer et al. (2019) examined urban heritage spaces in TX [164]. Seyfi et al. (2019) analyzed the factors that influence TX in cultural tourism [165]. Sanz-Blas et al. (2019) studied the relationships between destination image/familiarity, satisfaction, and behavioral intention for cruise tourists [166]. Olson and Reddy-Best (2019) studied TX for transgender and gender nonconforming individuals [84]. Sedgley et al. (2017) explored the experiences of mothers of children with autism spectrum disorder (ASD) during holidays [83]. Rubio–Escuderos et al. (2021) analyzed the TX of people with reduced mobility [167]. Van der Zee et al. (2017) studied the governance of tourism networks based on in-depth interviews with network managers [92].

Humagain and Singleton (2021) used semi-structured online focus group discussions to identify tourists' motivations, constraints, and negotiation strategies to participate in outdoor recreation trips during the COVID-19 outbreak [168]. Their study was re-printed 2 years later [169]. It is the only study based on a focus group.

Several netnography studies are based on data collected from TripAdvisor. Sthapit (2017) explored the MTX in hotels from Finland [170]. Thanh and Kirova (2018) studied the wine TX in Cognac, France [171]. Liu et al. (2019) analyzed the reviews made by foreign tourists that visited Beijing, China [172]. Oliveira et al. (2019) think more research on TX should be conducted based on online reviews, and they analyzed opinions on two islands of Cape Verde [173]. Yu et al. (2021) studied the city TX of London's top ten most popular tourist attractions [174]. Bigne et al. (2020) analyzed opinions on Spanish national parks and compared MTX vs. OTX dimensions [48]. Tokarchuk et al. (2022) analyzed the tourism carrying capacity in Berlin [175]. Yigit (2022) studied foreign tourists' cooking class experiences in Istanbul [176].

Bosangit (2015) indicated that travel blogs can offer insights into how tourists express the transformational effects of their experiences for the self [177]. Their study is based on the travel blogs of British bloggers. Anaya and Lehto (2020) examined the impact of technologies on TX, based on travel blogs from the last decades extracted from Travelblog.org [178]. Kim et al. (2020) studied the causes of negative TX based on Chinese travelogues available on Mafengwo [179]. Lin et al. (2022) studied TX in a National Wetland Park in China based on traveler reviews extracted from three popular Chinese online tourism communities [111].

Yang et al. (2021) focused on geo-tagged check-in user-generated content data, evaluating tourists' emotional experience as expressed on Chinese check-in pages on social media, through semantic analysis [180]. Liu et al. (2023) studied stranded travelers during the COVID-19 pandemic outbreak, focusing on post-traumatic growth [181]. They analyzed several resources: (1) chat records from seven WeChat mutual assistance groups of stranded travelers from Hubei Province, China,;(2) text data published by stranded travelers on online platforms; (3) information related to the stranded travelers reported by three Chinese news websites; and (4) official information released by the Chinese government.

Strijbosch et al. (2021) used electrophysiological measures and experience reconstruction in a musical theater show in the Netherlands [182]. They recorded physiological data with wearable wristbands (during-experience). Mitas et al. (2020) studied the relationship between emotional arousal and intent to recommend of visitors to Vincentre Museum and guided village tour in Nuenen, the Netherlands [183]. They measured emotional arousal as skin conductance responses. Bastiaansen et al. (2020) used psychophysiological measurements during a roller-coaster ride in Europapark, Germany [184].

Adongo et al. (2017) examined the relationship between TX and the intended length of stay in Ghana [185]. They used a self-reporting approach based on a survey with open-ended questions instead of scales. Coelho et al. (2018) focused on the core processes of MTX in an exploratory and qualitative approach based on the travel narratives of Brazilian

tourists [50]. When studying the transformative learning nature of Malaysian homestay experiences, Inversini et al. (2022) asked participants the self-record short videos answering a single broad open question: 'What have you learned or gained from this homestay experience?' [69]. Sørensen and Jensen (2016) conducted a field experiment in a retro-design boutique hotel in Copenhagen, in which service encounters were developed into experience encounters [186]. Moscardo (2020) explored the use of stories in designing dimensions of tourism and tourist experience [187].

Some studies were based on mixed TX evaluation methods. Servidio and Ruffolo (2016) focused their study on the relationship between MTX, emotions, and narratives, using a mixed quantitative and qualitative approach [46]. Participants were asked to identify the occurrence and describe six basic emotions at four stages of their travel: anticipation, travel to, on-site, and travel back. Sugathan and Ranjan (2019) performed three independent vignette-based experiments to manipulate levels of experience and co-creation [39]. Fang et al. (2023) examined the impact of word-of-mouth communication on TX storytellers, comparing oral and written communication [188]. Gardiner and Scott (2018) explored the development of new tourism experiences (TXs) on the Gold Coast, Australia [189]. They used a scale to assess the likelihood of tourists' participation in new TXs. Quantitative data were used for the planning of four focus group sessions. In their study of tourist segmentation in coastal tourism in West Ireland, Pafi et al. (2020) used semi-structured interviews in the first stage of their work [87]. In the second stage, they used a 16-item scale; the questionnaire also included an open-ended question. Gardiner et al. (2023) explore the role of self-identity in motivating participation in adventure tourism, comparing the views of youth from 4 countries (Australia, China, Singapore, and Germany) towards a learn-to-surf lesson [190]. Firstly, they conducted an in-depth mystery shopper research. In the second phase, an online survey was conducted. Dickinson et al. (2016) studied the TX of mobile disconnection in UK [191]. They used in-depth exploratory interviews and a 6-item scale. Moyle et al. (2017) assessed the preferences of potential visitors for nature-based experiences in protected areas in Australia through semi-structured interviews with stakeholders, and an online survey [192]. Gardiner et al. (2022) explored the staged authenticity in historical heritage tourism experiences in Australia, focusing on TX that has little relevance to where it was built [108]. Their study included semi-structured interviews, and a 22-item scale. In their study on tourism as an (integrative) service, Mendes et al. (2022) used focus groups in the first stage of their study, then a survey based on a scale in Portugal [40]. Komppula et al. (2016) examined TX in social holidays in Finland based on observations, semi-structured interviews, and questionnaires with open-ended questions [119]. When studying the engagement in diaspora tourism in mainland China, Weaver et al. (2017) used several qualitative methods: participant observation, casual discourse, auto-ethnography, and blog posts [76]. Gundersen and Rybråten (2022) analyzed how tourists, residents, and local stakeholders experience and practice their participation in the landscapes, based on focus groups, semi-structured interviews, an on-site survey, and internet surveys [193].

Hosseini et al. (2021) indicated that most of the studies on MTX are based on quantitative methods (52%); 36% of the studies are based on qualitative methods, and 12% are using mixed quantitative/qualitative approach [45]. In their review of MTX, Hosany et al. (2022) highlighted that studies usually employ quantitative methods and generally neglect negative experiences in TX [194]. Godovykh and Tasci (2020) indicated that most scholars do not evaluate TX as a whole; they focused on the cognitive and conative experience components based on self-reported pleasure from past experiences [38]. They highlighted several limitations of self-reported scales. They recommend evaluating TX moment-by-moment as it occurs, combining methods such as self-report scales, experience sampling, laboratory experiments, and psychophysiological techniques; various methods could maximize the TX insights. Ingram et al. (2017) acknowledged the need for improved TX evaluation methods and approaches [195]. They proposed the PART (prospective, active and reflective triangulation) as a "novel methodology for acquiring data before, during,

and after the holiday". Amaral et al. (2020) proposed a model for the design and evaluation of TX in seniors called OEC (organic, experiential, complex) [196].

Table 7 synthesizes the TX evaluation methods used in the studies we reviewed.

**Table 7.** TX evaluation methods.

| Method | Studies |
|---|---|
| Scale | [43,44,47,59,66,70,71,75,77,79,80,86,88,94–105,107,110,112,114–116,118,120,121,123–155] |
| Interview | [49,67,83,84,92,106,156–167] |
| Focus Group | [168,169] |
| Netnography | [48,111,170–181] |
| Physiological Measures | [182–184] |
| Other methods | [50,69,185–187] |
| Mixed methods | [39,40,46,76,87,108,119,188–193] |

Most of the TX studies were based on scales (54.9%), followed by interviews (14.8%) and netnography (11.5%). The most-used scale was that of MTX proposed by Kim et al. The scale was validated in cross-cultural contexts and was used by several authors in its initial adapted forms. Scales were included for a wide range of dimensions and items. Items were most frequently rated based on 5-point or 7-point Likert scale, but more detailed scales with up to 11 points were sometimes used. Remarkably, new scales were proposed instead of using scales proposed by other others. Scales need more validation in cross-cultural contexts. When evaluating TX in specific areas, generic scales may miss particular dimensions. Most of the studies focused on TX antecedents and consequences but not on TX evaluations. This may need to be clarified for TX practitioners.

Surveys based on scales are inexpensive and can identify TX weaknesses but not their causes. Scales should be complemented with other methods. There is a declared need for holistic TX evaluation, but few studies use mixed methods. Interviews are expensive, but netnography is relatively cheap. Most of the studies based on netnography use opinions collected from TripAdvisor.

Two TX evaluation methodologies (evaluation models) have been proposed; they are rather general. TX is constructed through several touchpoints, and holistic TX evaluations should focus on all (or at least the most relevant) touchpoints, dimensions, and context. If we consider TX to be an extension of UX focused on using tourism-related products, systems, and services, then UX evaluation methods may also offer valuable outcomes.

*5.4. RQ4: How Is the Post-Pandemic TX?*

Humagain and Singleton (2021) acknowledge that the COVID-19 outbreak has influenced tourists' psychology, behavior, and decision-making when participating in outdoor activities. They identify the lack of centralized and reliable information as a significant constraint when planning those activities [168]. Yang et al. (2021) indicate that tourists' (real-time on-site) emotional experience after the outbreak of COVID-19 was significantly lower than that pre-pandemic [180]. In their view, managers should focus on the post-pandemic TX. Kim et al. (2021) emphasize that mobility was stressful, and tourists' health was threatened during the COVID-19 pandemic [103]. Zhang et al. (2021) examine how the COVID-19 pandemic has changed social interaction in international tourism, and it could likely impact tourists' social identity and future behaviors [161].

Zhu et al. (2022) have examined the impact of theme park images on tourist-perceived value and behavioral intention during the COVID-19 pandemic [197]. Humagain and Singleton (2023) have studied tourists' motivations, constraints, and negotiations regarding outdoor recreation trips during the COVID-19 outbreak [169]. They indicate that COVID-19-related restrictions and fewer outdoor opportunities encouraged outdoor recreation, novelty-seeking, and experiencing normalcy. Buckley et al. (2022) argue that mental health research should be used in tourism [198].

Björk et al. (2020) highlight that in COVID-19 post-pandemic, "it is essential to examine how and to what extent the crisis changes peoples' fundamental values, motives, and tourist behaviors, how it boosts the short-distance/domestic market and virtual tourism experiences, and whether it may eventually benefit climate change adaptation and mitigation or provide solutions for over-tourism" [65]. They also indicate the need for research on sustainable and digital TX. Ioannides and Gyimóthy (2020) point out that the COVID-19 pandemic offers the opportunity to design and consolidate the transition towards greener and more balanced tourism [199]. Bosone and Nocca (2022)identify a growing awareness of sustainability and point out that this has been influenced by the health emergency of the COVID-19 pandemic [200]. Tokarchuk et al. (2022) highlight that, in the post-COVID-19 era, a negative TX due to overcrowding is a significant concern for destination managers [175]. Fusté-Forné (2023) has identified that slow tourism has been revalorized because of the COVID-19 pandemic [201]. Liu et al. (2023) reveal ten sources of trauma caused by the COVID-19 pandemic and faced by stranded travelers and identified five dimensions of post-traumatic tourism growth: meaning-seeking, reconstructing a life philosophy, identity, prosocial behavior, and awe of nature [181].

Altinay and Kozak (2021) acknowledge the possible shifts in tourist behavior and post-pandemic destination competitiveness [202]. They intend to "capture the edge of chaos of the tourism industry, the butterfly effects of COVID-19, cosmology, bifurcation events and behaviors, and health and safety-driven self-organization for destination competitiveness" through what they call "the butterfly competitiveness model". Magno and Cassia (2021) remark that the effects of tourism businesses' strategies to mitigate the COVID-19 crisis remain scarce [142]. They propose a model for the relationships between strategy and performance which includes five dimensions of corporate social responsibility behaviors.

Stankov and Gretzel (2020) highlight the importance of technologies in tourism, especially during the COVID-19 pandemic when virtual tourism was the only option [203]. They argue for the importance of using a human-centered design (HCD) approach in Tourism 4.0 technologies. Verma et al. (2022) recognize that virtual tourism emerged as an alternative to physical tourism due to the COVID-19 pandemic as a preview of the actual destinations and attractions [74]. Pasquinelli et al. (2023) study the online TX based on Airbnb Online Experience (AOE) reviews during the COVID-19 pandemic [204]. Their study highlights the role of the human dimension and human-to-human interaction, despite the digital mediation of the in-remote destination visit.

The COVID-19 pandemic was not only health-threatening but has undoubtedly influenced tourists' psychology, behavior, and decision-making. Emotional experience was remarkably different compared to the pre-pandemic era. Mobility was particularly stressful, social interaction was avoided, and tourists preferred outdoor recreation. There is an increasing awareness of the need for transitioning to greener, sustainable, more balanced, "slow" tourism and for corporate social responsibility behavior. Post-pandemic overcrowding was detected as a significant concern for destination managers.

As virtual tourism was at one point the only option, it became evident that technologies can and should play an increasing role in the post-pandemic era. Virtual tourism should not replace but should rather empower physical tourism. Human-centered design (HCD)'s fundamental approach in HCI and U was also highlighted as a must in TX design and management.

## 6. Conclusions

Tourism was dramatically affected by the COVID-19 pandemic, but it is now recovering. However, post-pandemic tourism should deal with new challenges. Overcrowding was detected as a significant concern for destination managers. Several authors have pointed out the increasing awareness of the need for transitioning to greener, sustainable, more balanced, "slow" tourism, as well as for corporate social responsibility behavior. Virtual tourism, the only option during the pandemic, will not replace but rather empower physical tourism, and technologies may help improve TX. Even if COVID-19 is no longer a

global emergency, lessons learned during the pandemic may be helpful when dealing with future challenges and crises. Human-centered design, a fundamental approach in HCI and UX, was also highlighted as necessary in TX design and management.

TX has been discussed in many studies. However, there still needs to be more consensus on its definition, dimensions, and evaluation methods. In our view, TX is a particular type of CX in which tourists are a specific case of customers interacting with tourism "brands" (organizations, companies). We consider CX and TX as extensions of UX. From a UX point of view, a tourist is a user of specific, tourism-related products, systems, or services. Based on the Godovykh and Tasci definition, we consider TX to be the cognitive, affective, sensory, and conative subjective perceptions, either negative or positive, and affected by situational factors, that a tourist has when interacting with brands, pre-, during, and post-travel, including their outcomes.

As several scholars indicate, TX is generated through a process that includes three stages (pre-, during, and post-travel), and the whole process should be analyzed. Most studies focus on MTX, but research should also address negative and ordinary TX. Many authors agree that TX is more than service quality and satisfaction, and a holistic, human-centered approach to TX is necessary. In order to properly understand and manage TX, its dimensions should be clearly identified. Most of the studies are based either on the seven-dimensional TX model of Kim et al. or on the experiential economy four-dimensional model of Pine and Gilmore (4Es). Godovykh and Tasci correctly identify four main TX dimensions: affective, cognitive, conative, and sensorial.

TX is usually evaluated in a quantitative approach, mainly based on scales. The MTX scale proposed by Kim et al. is used in its original or adapted form in many studies. TX scales include a wide range of dimensions and items. Items are usually rated on 5-point or 7-point Likert scales, but up to 11-point scales are sometimes used. Studies focus on examining the relationship between TX dimensions, its antecedents, and consequences rather than on TX evaluation results. Several scholars emphasize the importance of qualitative research in TX and the need for holistic TX evaluation, but only some studies use mixed methods. The need for standardized evaluation instruments may be problematic for TX practitioners. A holistic approach to TX should identify all touchpoints that occur in pre-, during, and post-travel stages; TX evaluation should explicitly focus on touchpoints, dimensions, and context of interest.

The main limitation of our study is that we have focused on only two scientific databases: Web of Science (WOS) and Science Direct (SD). We decided to do so in order to keep the information manageable. However, we think that this should not be a major concern, as most of the studies indexed in WOS and SD were published in high quality journals. We consider that the 167 articles that we fully analyzed cover the topics of our research well.

Only two TX evaluation models have been proposed, but they are rather general. The need for standardized, cross-culturally validated evaluation tools with both generic and specific evaluation models presents gaps that are still to be filled. If we consider TX an extension of UX, then UX evaluation methods offer valuable outcomes. Over the years, we developed several heuristics and evaluation models for UX tourism-related digital products. We also developed heuristics to evaluate CX in retail. In future work, we will particularly focus on developing TX heuristics and checklists.

Our research contributes to better understanding the rather heterogenous approaches to TX. We propose a holistic TX definition that considers TX as a UX extension and a particular case of CX. Our study identifies the trends in TX research over the last decade. It also highlights several gaps: (1) the relative lack of qualitative and mixed TX research, (2) the neglect of ordinary and negative TX, and (3) the scarcity of TX (evaluation) models. Our study has both theoretical and practical contributions. It certainly helps researchers to identify relevant TX definitions, dimensions, and models, and the research gaps to be addressed in the future. It may also help TX managers to select appropriate TX design and evaluation tools, both general and specific.

Our study highlights several future research directions. First, there is still a need for TX definitions. The holistic definition that we propose can certainly be improved. Second, comprehensive TX models are still necessary. General and specific TX dimensions and subdimensions (factors, attributes) must be clearly identified, as they are establishing the ground for both TX evaluation and design. Third, TX evaluation models (methodologies) are yet to be established. They should allow for the evaluation of specific TX aspects as well as holistic TX evaluations.

**Author Contributions:** Conceptualization, V.R. and C.R.; methodology, V.R. and C.R.; validation, V.R., C.R., N.M. and F.B.; formal analysis, V.R., C.R. and N.M.; investigation, V.R., C.R., N.M. and F.B.; resources, V.R. and C.R.; data curation, V.R., C.R. and N.M.; writing—original draft preparation, V.R. and C.R.; writing—review and editing, V.R., C.R., N.M. and F.B.; visualization, V.R., C.R., N.M. and F.B.; supervision, V.R. and C.R.; project administration, V.R.; funding acquisition, V.R. and C.R. All authors have read and agreed to the published version of the manuscript.

**Funding:** This research was funded by Dirección General de Investigación of Universidad de Playa Ancha de Ciencias de la Educación, Chile, Concurso Regular 2020, code HUM 04-2122. The APC was funded by Pontificia Universidad Católica de Valparaíso, Chile.

**Institutional Review Board Statement:** Not applicable.

**Informed Consent Statement:** Not applicable.

**Data Availability Statement:** All data generated or analyzed during this study are included in this article.

**Acknowledgments:** Nicolás Matus is a beneficiary of ANID-PFCHA/Doctorado Nacional/2023-21230171.

**Conflicts of Interest:** The authors declare no conflict of interest.

## Appendix A

**Table A1.** TX scales.

| Study | Area | Context | Dimensions | Items and Rating |
|-------|------|---------|------------|------------------|
| [43,123] | MTX | USA, Taiwan | Hedonism, Refreshment, Local culture, Meaningfulness, Knowledge, Involvement, Novelty | 24 items, 7-point Likert scale |
| [44] | MTX | Taiwan | Local culture, Variety of activities, Hospitality, Infrastructure, Environment management, Accessibility, Quality of service, Physiography, Place attachment, Superstructure | 33 items, 7-point Likert scale |
| [124] | Hotels | Spain | 7 dimensions from Kim et al. 2012, Destination image, Tourist engagement, Satisfaction, Revisit intention, Recommendation intention | 48 items, 7-point Likert scale |
| [112] | Rural TX | China | 7 dimensions from Kim et al. 2012, Education, Esthetic, Entertainment, Escapism, Connectedness to nature, Environmental awareness, Green consumption | 61 items, 7-point Likert scale |
| [125] | Zoo and museum | Finland | 4 dimensions from Kim et al. 2012 | 30 items, 5-point Likert scale |
| [126] | Destination | Finland | 7 dimensions from Kim et al. 2012 | 30 items, 5-point Likert scale |
| [127] | MTX | USA | 7 dimensions from Kim et al. 2012 | 42 items, 5-point frequency scale |

**Table A1.** *Cont.*

| Study | Area | Context | Dimensions | Items and Rating |
|---|---|---|---|---|
| [128] | Eco-tourism | Iran | 7 dimensions from Kim et al. 2012 | 33 items, 5-point Likert scale |
| [47] | MTX | China | 7 dimensions from Kim et al. 2012 | 60 items, 5-point Likert scale |
| [129] | Sardinia | Italy | 4 dimensions from Kim et al. 2012 | 39 items, 5-point Likert scale |
| [130] | Natural Attractions | Croatia | 3 dimensions from Kim et al. 2012 | 39 items, 5-point Likert scale |
| [94] | MTX | China | 3 dimensions from Kim et al. 2012 | 31 items, 7-point Likert scale |
| [131] | Destination | Iran | 3 dimensions from Kim et al. 2012 | 35 items, 5-point Likert scale |
| [132] | MTX | Turkey | 6 dimensions from Kim et al. 2012 | 20-items, 5-point Likert scale |
| [133] | MTX | Brazil | Environment, Culture, Relationship with companions, Relationship with tourists, Relationship with local agents, Dream, Emotion, Novelty, Refreshment, Meaningfulness | 41 items, 7-point Likert scale |
| [114] | Island | South Korea | Escapism, Relaxation, Enjoyment, Involvement, Perceived value, Perceived price reasonableness, Satisfaction with tour experience, Loyalty to an island, Island image | 30 items, 7-point Likert scale |
| [97] | City | Spain | Cognitive, Affective, Delight | 35 items, 5-point Likert scale |
| [134] | MTX | Portugal | Senses, Emotions, and Memories | 47 items, 7-point Likert scale |
| [135] | Sensory stimuli | Spain | Visualscape, Smellscape, Tastescape, Soundscape, Hapticscape | 17 items, 7-point Likert scale |
| [136] | TX Touchpoints | China | 4 types of Touchpoints (destination-owned, partner-owned, tourist-owned, and social) and 2 categories of Well-being (hedonic and eudaimonic) | 32 items |
| [96] | Well-being | China | Sensory, Feelings, Knowledge, Behavior, authenticity, Co-creation, Novelty, Recollection, Storytelling, Well-being | 39 items, 7-point Likert scale |
| [137] | Negative tourist-to-tourist interaction | Ghana | Interpersonal directed negative interaction, Interpersonal non-directed negative interaction, Site-directed negative interaction, Intrapersonal negative interaction, Value, Memorability | 26 items, 5-point Likert scale |
| [77] | Negative tourist-to-tourist interaction | Turkey | Interpersonal directed negative interactions, Interpersonal non-directed negative interactions, Site-directed negative interactions, and Intrapersonal negative interactions | 51 items, 7-point Likert scale |
| [104] | Island | Spain | Social stimulation, Privacy level, Behavioral constraints, Perceived control, Social setting characteristics, Physical setting characteristics, Personal characteristics | 34 items, 9-point scale |
| [138] | Cruise | China | Environmental stimuli and TX | 16 items, 5-point Likert scale |
| [75] | Cruise | USA | Social interaction, Interaction with employees, Relatedness, Activity-related tourist engagement | 16 items, 5-point Likert scale |
| [59] | City | Taiwan | Experiential stimuli, Flow, Positive emotion, Perceived value, Satisfaction, Intention to recommend, Intention to revisit | 24 items, 7-point Likert scale |

**Table A1.** *Cont.*

| Study | Area | Context | Dimensions | Items and Rating |
|-------|------|---------|------------|------------------|
| [99] | City | Hungary | Interaction, Interactive experience environment, Organizational experience environment, Customization, Experience involvement, Experience authenticity, Memorability | 43 items, 7-point Likert scale |
| [139] | Cultural | Portugal | City attributes | 18 items, 5-point Likert scale |
| [140] | City | Spain | Information, Entertainment, Self-Expression, Satisfaction, Intention to visit a City, Mobile Convenience, TX | 27 items |
| [100] | Cultural | China | Sensory experience, Action experience, Emotional experience, Thinking experience | 15 items, 7-point Likert scale |
| [110] | Urban forest | China | Destination image, Recreational benefits, Satisfaction, Perceived value, Behavioral intentions | 40 items, 5-point Likert scale |
| [141] | Urban wintersport | Austria | Push factors, Pull factors, Satisfaction, Cultural satisfaction, Winter sports, Intention to revisit | 42-items, 5-point Likert scale |
| [118] | Agrotourism | Indonesia | Uniqueness, Learning, Staff, Escape, Peace-of-mind | 31 items, 5-point Likert scale |
| [142] | Agrotourism | Italy | Community, Employees, Environment, Heritage, Products, Proactive strategies, Reactive strategies, Co-creation experience, Performance | 32 items, 5-point Likert scale |
| [116] | Rural | South Korea | Destination quality, Family cohesion, Refreshment, Novelty, Arousal, Meaningfulness, Involvement | 31 items, 7-point Likert scale |
| [143] | Ecotourism | Dominican Republic | Ecotourism knowledge, Attitudes towards ecotourism, Perceived value of ecotourist site, Ecotourist satisfaction, Behavioral intentions | 32 items, 5-point Likert scale |
| [95] | Environmental sustainability | Spain | Cognitive, Affective, Behavioral, Sensorial, Social, Environmental sustainability, Tourist satisfaction | 36 items, 5-point Likert scale |
| [115] | Low-carbon tourism | Taiwan | Sensory, Affective, Learning, Sociocultural, Behavioral, Escapism, Prestige | 40 items, 7-point Likert scale |
| [144] | Wetland parks | China | Post-trip Behavioral Intention, Action Experience, Aesthetic Experience, Emotional Experience, Resource Conditions, Recreational Activities, Tourism Facilities, Integrated Management, Related Personnel | 43 items, 5-point Likert scale |
| [145] | Wetland parks | China | Importance and pleasure, Sign value, Risk probability and consequence, Resource and environment experience, Facility management experience, Environmentally responsible behavior | 23 items, 5-point Likert scale |
| [86] | Antartic | Antartic | Perceived learning, Measured learning, Satisfaction, Environmental concerns, Management preferences, Behavior intentions | 38 items, 11-point Likert scale, True/False Quiz |

**Table A1.** *Cont.*

| Study | Area | Context | Dimensions | Items and Rating |
|---|---|---|---|---|
| [153] | Coastal | Taiwan | Destination image, Constraint, Escapism, Entertainment, Esthetics, Education | 45 items, 5-point Likert scale |
| [146] | Rural and coastal | Spain | MTX, Destination sustainability | 19 items, 5-point Likert scale |
| [147] | Coastal | Denmark, UK | Satisfying marina destination experiences | 28 items, 5-option scale |
| [98] | Religious | Taiwan | Relaxation benefit, Health benefit, Brand image, Culture learning, Peace of mind, Escape, Unique involvement, Interactive opportunities | 29 items, 7-point Likert scale |
| [70] | Religious | Taiwan | Spirituality, Learning, Physicality, Help, Unpleasantness | 19 items, 7-point Likert scale |
| [88] | Halal | Indonesia | Holistic experience, Religiosity, Satisfaction, Intention to revisit, Intention to endorse | 41 items, 5-point Likert scale |
| [107] | Industries parks | Taiwan | Learning, Recreation, Exhibitions, Service, Food, Facilities, and Souvenirs | 27 items, 7-point Likert scale |
| [148] | Wine regions | Several Countries | Emotional attachments to place-based brands | 7-point Likert, semantic differential, and metric scale |
| [101] | Entertainment | Macau | Learning, Enjoyment, Escape, Refreshment, Novelty, Involvement, Local culture | 26 items, 7-Likert scale |
| [120] | Hallmark festivals | Australia | Emotional value, Functional value, Monetary value, Social approval value, Self-image congruency value, Safety value, Tourist Intentions, Wellbeing, Festival Attachment | 29 items, 7-point Likert scale |
| [66] | Restaurant | Turkey | Food, Service, Atmosphere, Authenticity, Destination food image | 29 items, 5-point Likert scale |
| [80] | Smartphone and TX | Taiwan | Travel motivation, TX, Satisfaction, Sharing motivation, Smartphone usage, Pastime | 56-items, 5-point Likert scale |
| [149] | Nature-based | Taiwan | Aesthetics, VR/AR presence, Usefulness, Ease of use, Hedonic experience, Trust, Learning experience, Satisfaction, Loyalty | 30-items, 7-point Likert scale |
| [150] | Digital information quality | Thailand | Destination image, Perception of destination attributes, Satisfaction, Behavioral intention | 34-items, 5-point Likert scale |
| [103] | Mobility app | South Korea | Usefulness, Trust, Interactivity, Behavioral control over mobility, Stress from mobility, MTX through mobility app, Mobility app reuse intention | 32-items, 7-point Likert scale |
| [105] | International travelers | Iran | Place identity, Place attachment, Place dependence, Perceived quality of and value of attractions, Satisfaction | 23-items, 5-point Likert scale |
| [151] | Regenerative tourism | Saudi Arabia | Escapism, Relaxation, Enjoyment, Involvement, Perceived price reasonableness, Regenerative tourism involvement, Destination loyalty, Destination image, Satisfaction | 38 items, 7-point scale |

**Table A1.** *Cont.*

| Study | Area | Context | Dimensions | Items and Rating |
|---|---|---|---|---|
| [71] | Destination loyalty | China | Pre-trip involvement, Satisfaction with co-creation, Place attachment, Destination loyalty | 17-items, 7-point Likert scale |
| [152] | Airport loyalty | Brazil | Expectation, Perceived value, Passenger satisfaction, Image, Complaints, Switching costs, Loyalty, Check-in, Security, Convenience, Ambiance, Basic facilities, Mobility | 59-items, 7-point Likert scale |
| [102] | Creative TX | Taiwan | Uniqueness, Identification, Attractiveness, Perceived coolness, Satisfaction, Place attachment, Destination loyalty | 34-items, 5-point Likert scale |
| [121] | Seniors | Australia | Self-determined motivations, Tour preferences, Memorable experiences, Feelings, Perceived benefits, Life satisfaction | 78 items, 7-point and 5-point Likert scales |
| [154] | TX | USA | Reverie, Tourismscape, Contemplation, Transformation, Amusement | 56 items, 7-point Likert scale |
| [155] | Safari Park | Denmark | Experiential expectations | 3 questionnaires, 7-point Likert scale |
| [79] | Family | Japan | Positive emotion, Engagement, Relationship, Meaning, Accomplishment | 11-point scale, 5-point scale, 5-emoji scale for children |

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
