# Peer review of "Tourist Experience Challenges: A Holistic Approach"

_sustainability, doi:10.3390/su151712765_

Round 1

Reviewer 1 Report

This article presents an analysis of 167 articles on the travel experience and the impact of the pandemic on it.

The article meaningfully reveals the set tasks. The methods correspond to the research task. The logical structure and the arguments make it possible to understand the author's position.

With all the strengths of the article, the authors need to link the title and the content. In the Introduction, it is necessary to show that the author's approach is holistic. This will emphasize the novelty and relevance of the work.

Author Response

Dear Editors and Referees,

We really appreciate your feedback. As instructed, we revised the manuscript according to all your very helpful comments. We hope the updated version of the manuscript has been improved.

Best regards,

Virginica Rusu, Cristian Rusu, Nicolás Matus, and Federico Botella

Response to Reviewer 1 Comments

Comment 1: This article presents an analysis of 167 articles on the travel experience and the impact of the pandemic on it.

The article meaningfully reveals the set tasks. The methods correspond to the research task. The logical structure and the arguments make it possible to understand the author's position.

Response 1: We really appreciate your kind comments.

Comment 2: With all the strengths of the article, the authors need to link the title and the content. In the Introduction, it is necessary to show that the author's approach is holistic. This will emphasize the novelty and relevance of the work.

Response 2: Thank you for your suggestion. Introduction and conclusions were adapted accordingly.

Response to Reviewer 2 Comments

Comment 1: First of all I would like to thank you very much for such important article.

In the article Tourist Experience Challenges: A Holistic Approach the authors propose a holistic definition of Tourism eXperience (TX) and set recommendations for better analysis with using TX idea.

Paper is clear and relevant for the field of study and it is presented in a well-structured manner. The cited literature is recent and important for the subject of the research. The manuscript is maintained in a structure that meets the requirements of a scientific article. Presented analysis are reproducible and the description of the method is adequate. All the tables and images are appropriate and they properly show the data. Tables and images are easy to interpret and understand. All source information are interpreted appropriately and consistently throughout the manuscript. Conclusions are consistent with the evidence and arguments presented. Research questions are correct.

The article is interesting and well written. The text is clear and easy to read.

The strengths of the article are a deep literature analysis and consideration of a very interesting and important research area.

Response 1: We really appreciate your kind comments.

Comment 2: The article lacks an indication of the limitations of the conducted research.

Response 2: We appreciate your suggestion. The limitations of the study are now stated in conclusions.

Comment 3: At the end of the article, it is always worth pointing out further research directions.

Response 3: Thank you for your suggestion. The updated version of the manuscript indicates future research directions in conclusions.

Author Response

(The authors gave the same response as above.)
